# Clinical Nutrition in Portuguese Gastroenterology Departments: A Multicentric Study

**DOI:** 10.3390/ijerph192316333

**Published:** 2022-12-06

**Authors:** Francisco Vara-Luiz, Luísa Glória, Irina Mocanu, António Curado, Isabel Medeiros, Maria Antónia Duarte, António Banhudo, Susana Ferreira, Ana Margarida Vaz, Isabel Bastos, Jorge Fonseca

**Affiliations:** 1GENE–Artificial Feeding Team, Gastroenterology Department, Hospital Garcia de Orta, 2805-267 Almada, Portugal; 2Grupo de Patologia Médica, Nutrição e Exercício Clínico (PaMNEC), Centro de Investigação Interdisciplinar Egas Moniz (CiiEM), 2829-511 Almada, Portugal; 3Gastroenterology Department, Hospital Beatriz Ângelo, 2674-514 Loures, Portugal; 4Gastroenterology Department, Centro Hospitalar do Oeste, 2500-176 Caldas da Rainha, Portugal; 5Gastroenterology Department, Hospital do Espírito Santo, 7000-811 Évora, Portugal; 6Gastroenterology Department, Hospital do Divino Espírito Santo de Ponta Delgada, 9500-370 Ponta Delgada, Portugal; 7Gastroenterology Department, Hospital Amato Lusitano, 6000-085 Castelo Branco, Portugal; 8Gastroenterology Department, Hospital do SAMS, 1849-017 Lisboa, Portugal; 9Gastroenterology Department, Centro Hospitalar Universitário do Algarve, 8000-386 Faro, Portugal; 10Gastroenterology Department, Centro Hospitalar do Baixo Vouga, 3810-164 Aveiro, Portugal

**Keywords:** nutrition, training, nutrition techniques, gastroenterology

## Abstract

Background: Hospital nutrition is a major public health problem, as up to 50% of hospitalized patients suffer from undernutrition. Adequate nutritional support (NS) decreases morbidity/mortality, shortens the length of stay, and reduces costs. We aimed to evaluate the engagement of Portuguese gastroenterology departments in NS, especially in artificial nutrition (AN). Methods: Cross-sectional multicentric study, using an online survey sent to 31 Portuguese gastroenterology departments. Results: Nine centers were involved, and all departments were engaged in NS activities. The most performed nutrition technique was endoscopic gastrostomy and not all departments had the expertise to perform all nutrition procedures, namely, endoscopic jejunostomy. Two departments had an AN outpatient clinic. Five centers were involved in hospital nutrition committees. Only four performed systematic nutritional evaluation of every patient on admission. Two departments developed research in the nutrition field. An increase staff and nutrition training were pointed out as suggestions to improve NS. Conclusions: This study outlines a broad picture of NS/AN in Portuguese gastroenterology departments. Medical nutritional training and increasing nutrition teams’ staff may contribute to developing NS/AN. Multidisciplinary management of nutrition-related disorders is of utmost importance, and gastroenterologists are expected to be at the core of hospital nutrition.

## 1. Introduction

Nutrition plays a major role not only in everyday human life but also in clinical practice. Undernutrition is defined as a state resulting from a lack of intake or uptake of nutrition that leads to altered body composition, diminished physical and mental function, and impaired clinical outcome from disease [1]. Furthermore, it is estimated that up to 50% of hospitalized patients will suffer from undernutrition, which represents a major public health problem and, frequently, an overlooked one [2]. This phenomenon is also documented in Portugal, where more than 40% of hospital patients are undernourished or at risk of developing undernutrition at admission, and this panorama tends to worsen during the hospital stay [3].

According to the European Society for Clinical Nutrition and Metabolism, there are many forms of nutrition care and therapy: regular meals, personalized diets, medical nutrition therapy and palliative nutrition [4]. Nutritional support (NS) may be used with both a supportive and therapeutic role in the management of a wide range of disorders [5]. In hospitalized patients, NS has been shown to positively affect morbidity and mortality, shorten the length of stay, and reduce health-care costs [6,7].

Despite the importance of nutritional knowledge for health professionals, such education has been scarce at both undergraduate and postgraduate levels [8]. Several studies among general practitioners and hospital doctors report dissatisfaction with their postgraduate training in nutrition [9,10], as well as poor training in assessment and management of undernutrition [11,12]. Although medical doctors are responsible for prescribing NS, several studies identify discrepancies between evidence-based nutritional guidelines and actual clinical practice [13,14].

The gastroenterology departments have an important responsibility to provide advice on general hospital nutrition, especially on artificial nutrition (AN). Both in health and in disease, human nutrition results, along with other factors, from the complex interplay between the digestive system and the brain. Furthermore, gastroenterologists treat undernutrition associated with illness-related anorexia, chronic inflammation, gut obstruction, and malabsorption. For instance, undernutrition occurs in 20–50% of patients with chronic liver disease, and its progression is associated with a higher rate of complications and liver failure. Thus, NS may improve the prognosis of cirrhotic patients [15]. In addition, the interplay in the nutrition–inflammatory bowel disease is complex, as some patients identify associations between diet and disease flare, and nutrition intervention may be a valuable component of treatment [16]. In addition, some digestive disorders have mandatory nutritional therapy. As a standard, the best therapy for celiac disease is strict gluten avoidance [17], which represents a challenge due to its restrictive nature [18]. Moreover, gastroenterologists contribute with their expertise to treat undernutrition in patients with digestive tract obstruction or motility disorders by performing techniques such as prosthesis placement and percutaneous endoscopic gastrostomy (PEG)/jejunostomy (PEJ), providing an enteral nutrition route. In addition, they provide long term parenteral nutrition to patients with short bowel syndrome and other forms of type III and type II intestinal failure.

According to the European Board of Gastroenterology and Hepatology [19], gastroenterologists have the advantage of recognizing undernutrition or conditions that threaten patients’ nutritional status at an early stage, can visualize them, and then can intervene. Therefore, nutrition training is of utmost importance as a part of the gastroenterology training program and curriculum. In addition, given the major influence of the gastrointestinal system in human nutrition, gastroenterologists may take a leading role in the management, training, and research on nutrition, as nutrition is a major contributor to health and diseases of digestive and systemic disorders [20]. The aim of the present study was to evaluate the actual state of engagement of gastroenterology departments in nutrition activities across Portuguese hospitals. Specific aims included:Department characterization.Evaluation of departments engagement in medical training and in research.Characterization of inpatient nutrition interventions.Characterization of departments’ involvement in hospital nutritional counseling.Enteral and parenteral feeding techniques performed.Evaluation of the existence and characterization of outpatient nutrition clinics.Obtaining suggestions from the heads of the departments for developments in nutritional practice.

## 2. Materials and Methods

The authors performed a cross-sectional multicentric study based on an online survey addressed to the heads of gastroenterology departments belonging to the Núcleo de Gastrenterologia dos Hospitais Distritais (NGHD) that concerned their involvement in hospital-based NS. The NGHD is an organization that represents most Portuguese public gastroenterology departments, except for some university hospitals and specialized cancer hospitals. In total, 31 hospitals from all of Portugal’s regions are affiliated with the NGHD.

The questionnaire contained an introduction presenting the rationale behind the study and its aims. The questions covered several areas: 1. department characterization (population of the area of influence, ward facilities, types of outpatient clinic); 2. nutrition training and research; 3. inpatient nutritional interventions; 4. gastroenterology departments’ involvement in hospital’s nutritional counseling; 5. enteral and parenteral feeding techniques performed; 6. outpatient nutrition clinic; and 7. future development in the nutritional support field. The online survey was carried out using the Google Forms platform and was sent to the heads of all gastroenterology departments of Portuguese hospitals that are partners of the NGHD (*n* = 31). The answers were collected during from July to October 2019 for data from 2018. One reminder was sent to nonresponders in September. The authors considered nonresponders those who had not answered by 31 October 2019.

The data were inserted into a database and a descriptive statistical analysis was performed using Microsoft Office Excel, as well as linear correlations on Statistical Package for Social Sciences 28. Data are presented as frequencies for categorical variables and medians for continuous variables. In data presented in tables, the hospital’s names have been concealed.

## 3. Results

A total of nine gastroenterology departments’ responses were collected through the questionnaire, which represents a 29% response rate. All the included hospitals were involved in some form of artificial nutrition.

### 3.1. Department Characterization

Among the responders, there were two hospitals from the northern region of Portugal, one from the center, three from the Lisbon metropolitan area, two from the south, and one from the Azores. One of them was a university hospital center. All the included hospitals have an associated area of influence of 2,203,000 citizens, which represents 23% of the Portuguese population. In this regard, the ratio of patient per doctor ranges from 12,000 to 95,000.

Table 1 summarizes general characteristics concerning area of influence, ward facilities and number of gastroenterologists, as well as the existence of AN outpatient clinic. Four departments did not have specific ward facilities, although following inpatients in other departments’ wards. All of them worked with a dietitian shared with other hospital departments.

### 3.2. Nutrition Training and Research

Regarding nutrition training, in five of the departments (55.6%), gastroenterology trainees were enrolled in nutrition-related courses. As shown in Figure 1, most of the departments chose the Nutrition in Gastroenterology course, by the Portuguese Society of Gastroenterology.

Two departments have developed research in this area: posters and oral presentations in national and international meetings and papers in national and international journals. Members from one department had written a nutrition-related chapter of a scientific book.

### 3.3. Inpatient Nutrition Interventions

Concerning nutritional assessment in ward facilities, regarding the hospitals with inpatient facilities (*n* = 5), four of them (80%) perform nutritional risk screening and evaluation on admission. In half of the departments (*n* = 2), this work was performed by a nurse, and in the other half this was performed by a dietitian. During hospitalization, two of the departments admitted to not codifying undernutrition on every occasion for diagnosis-related administrative purposes.

During clinical grand rounds, nutritional issues were discussed in circa 50% of cases in two departments, 75% in another two, and for every patient (100%) in only one department.

### 3.4. Department Involvement in Hospital Nutritional Counseling

Concerning gastroenterology departments’ involvement in hospitals’ nutrition-related issues, three departments provide support to others regarding AN decisions, including participating in a multidisciplinary team. Furthermore, five departments (56%) are involved in the hospital nutrition committee.

### 3.5. Enteral and Parenteral Feeding Techniques Performed

All the included departments (*n* = 9) performed PEG placement, which was the most frequent long-term nutritional support route, followed by nasoenteric tube (NET; *n* = 8) and endoscopy-assisted nasogastric tube placement (NGT; *n* = 7), as depicted in Table 2. In contrast, none of the departments had performed percutaneous endoscopic jejunostomy (PEJ) during 2018. In cases where a department did not perform a procedure, they were asked whether they had experience and expertise on the procedure if necessary. All the departments had expertise in placing endoscopic NGT and NET, as opposed to PEG-J or PEJ. There is a moderate linear correlation (Pearson correlation coefficient = 0.612) between the number of doctors and expertise in nutrition techniques (Figure 2).

Regarding catheter placement for parenteral nutrition, the most frequently performed procedure was the tunneled central venous catheter, e.g., Hickman/Broviac catheter (*n* = 3) and the peripheral venous catheter (*n* = 3). Again, not all departments have the experience to perform those procedures if necessary.

### 3.6. Outpatient Nutrition Clinic

Only two departments from the responders have an artificial nutrition-related outpatient clinic. In both, patients were assessed in a nutritional team comprised of a gastroenterologist, a dietitian, and a nurse. Patients with enteral nutrition by gastrostomy/jejunostomy (either endoscopic or surgically performed) and with parenteral feeding are assessed at these clinics. Moreover, nutritional support appointments are available for patients with inflammatory bowel disease (IBD) and hepatic diseases in all departments that offer these outpatient clinics. Only one department follows patients with oral supplemented nutrition and oral adapted nutrition.

According to the responders, the most frequent pathologies that require ambulatory AN are neurologic disease, head and neck cancer, and intestinal failure.

### 3.7. Future Development in the Nutritional Support Area

Finally, when asked to point to any suggestions that would improve the nutritional support in their departments, the most frequent proposal (*n* = 3) was to increase the staff of the nutritional clinic. Other suggestions outlined were training of the teaching fellows in their department (*n* = 1), creating specific outpatient appointments regarding nutrition in specialist fields, e.g., IBD, chronic pancreatitis (*n* = 1), and a larger involvement in the ambulatory setting (*n* = 1).

## 4. Discussion

During the last few years, there has been increased interest in the impact of nutrition in disease and its importance in patient outcomes [5,6,7]. AN, either enteral or parenteral, is necessary when oral feeding is impossible, insufficient, or unsafe, but it is complex and comes with a wide variety of clinical risks. Well-controlled procedures and close monitoring of patients are essential and should be guaranteed by an experienced and multidisciplinary team [20,21,22]. Gastroenterologists are usually requested to be part of AN procedures and hospital nutritional support teams, and should be prepared for increasing demands of these nutrition activities.

The present study outlines a broad picture of the actual state of nutrition activities’ engagement of gastroenterology departments across Portuguese hospitals and is the first evaluating this matter in our country. The departments included had important differences regarding ward facilities, number of physicians, and types of outpatient clinic. As expected, those with fewer physicians were the ones without specific wards and with fewer types of outpatient appointments. Furthermore, we found a diverse although high ratio patients per doctor between the involved centers, which may adversely impact nutrition health care. All the departments included a dietitian shared with other clinical areas, which is important to assess and manage patients on admission and during hospitalization. Regrettably, even when identified, undernutrition found in inpatients was not always codified. This may become a burden, because a lack of codification results in a lack of reimbursement and identification of malnutrition as an important clinical problem. The implementation of simple routines such as nutrition documentation at admission, not performed by all departments, and predefined medical staff with specific responsibilities regarding common nutrition routines may help to overcome this problem and raise awareness of the importance of this matter in hospitalized patients.

Regarding nutrition training, only five departments (55.6%) have participated in nutrition-related courses, mainly but not exclusively the one organized by the Núcleo de Nutrição em Gastrenterologia, the special interest group of the Portuguese Society of Gastroenterology. This nutrition course became an important educational tool for Portuguese gastroenterologists and residents, but is just a first step. Standardized and structured core nutrition training would be of value to increase expertise among practitioners. The scarce information on nutrition is also identified in many European countries, where training programs for medical and surgical specialties lack formal nutrition support training/education [23], resulting in an incapacity to assess patients’ nutritional requirements properly [24,25,26]. In addition, only two departments developed research in nutrition, a low number given the recognized importance of this field and the potential impact on patient comorbidities and mortality [6].

Regarding endoscopic techniques for nutritional support, the present study found that all departments performed the most common ones (PEG, NGT, and NET) during 2018. In contrast, some departments stated that they had no experience in performing several techniques, such as PEG-J and PEJ, important in patients that do not tolerate long-term prepyloric enteral feeding. Without endoscopic expertise in these procedures, the other option would be surgical jejunostomy, which is significantly more costly and bears the risks of a surgical procedure. As such, the choice of the feeding route is complex and depends on the underlying pathology, expected duration of enteral nutrition, preference of the patient and ethical considerations, highlighting the importance of nutritional training in gastroenterology. Training and expertise improve patient outcomes, as seen in other centers, where the 30-day mortality after PEG insertion has declined 60% over 13 years [27]. In fact, nutrition knowledge and skills are so important that the National Institute for Clinical Excellence guidelines [28] states that, “all healthcare professionals who are directly involved in patient care should receive education and training…that should cover options for all nutrition support (oral, enteral and parenteral).” In fact, as different clinical scenarios may require different approaches (PEG, PEG-J, enteral stenting, biliary stenting, etc.), interventional endoscopy training can impact nutritional status and improve patients’ prognosis.

Although nutrition should be provided by every clinician, there is some expectation for gastroenterologists to have specific expertise, given the tight link between the digestive system and nutritional health [8]. Gastroenterologists should have the responsibility to be engaged in the management of disease-related undernutrition [14], but only two departments have artificial nutrition-related appointments. Moreover, all included departments deal with a very different spectrum of patients, each of them needing personalized nutrition care. For instance, nutritional status of cancer patients is of utmost importance, as some treatments may be impeded or precluded by the development of undernutrition [29]. For this reason, outpatient nutrition appointments should be provided by most of the gastroenterology departments, and that reality is still far away.

When asked, the heads of the departments pointed out an increment in medical and nonmedical staff as the main step to develop the nutrition care of their patients. The reduced staff might explain the low percentage of nutrition-related appointments and low number of performed techniques, the last demonstrated by a linear correlation in our study.

Also, the implementation of a hospital multidisciplinary team regarding nutrition issues, composed of several departments, including gastroenterology, is mandatory to improve nutrition care of patients in ward facilities and the ambulatory setting. Another suggestion given was training for the teaching fellows in their departments, an essential component of providing adequate NS. However, not only the fellows should receive nutritional training but the entire medical staff, in order to share knowledge and increase expertise.

A major strength of this study was its multicentric nature, involving departments of different characteristics and from different regions across the country. However, some limitations need to be outlined. First, this survey only included partners of NGHD, excluding some larger hospitals that may have different experience related to AN. Moreover, despite sending this survey to 31 departments, only nine replied (29% response rate). However, the study included very different gastroenterology departments, from small units to large academic centers, creating a broad spectrum of observed centers. Moreover, the area of influence of all involved centers makes up 23% of Portuguese population, more than 2.3 million citizens. Thus, the authors believe this study may be representative of the clinical settings regarding most of the national gastroenterology departments. Secondly, this survey was delivered only to the head of each department. Although their perspective may represent a more comprehensive view, it may differ from the other colleagues, including fellows. Finally, this survey regarded data from 2018 collected in 2019 and thus does not cover the last few years or the impact of COVID-19 on gastroenterology practice. Nevertheless, it is reasonable to assume that the 2020–2021 efforts of the gastroenterology departments were focused on dealing with COVID-19, and significant advances in nutrition practices from the answers given in 2019 would not be expected.

In the coming years, a new study evaluating advances in nutrition care, data regarding enteral stenting, endoscopic retrograde cholangiopancreatography and endoscopic ultrasound for nutrition purposes, as well as the impact of COVID-19 pandemic on nutrition is warranted, hopefully with the involvement of more centers across the country. Gastroenterology departments must increase their commitment to clinical nutrition and artificial nutrition.

## 5. Conclusions

Our study supports the idea that gastroenterology has an important role in nutrition, as all included hospitals were involved in some form of AN and had technical expertise for the most common procedures if needed. Nevertheless, a great deal of work remains to be done in this area. Nutrition training for gastroenterologists is essential but long overdue, and strategies for its development should be implemented. Multidisciplinary management of nutrition-related disorders is of utmost importance to deal with hospital malnutrition, and gastroenterologists must be at the core of this practice.

## Figures and Tables

**Figure 1 ijerph-19-16333-f001:**
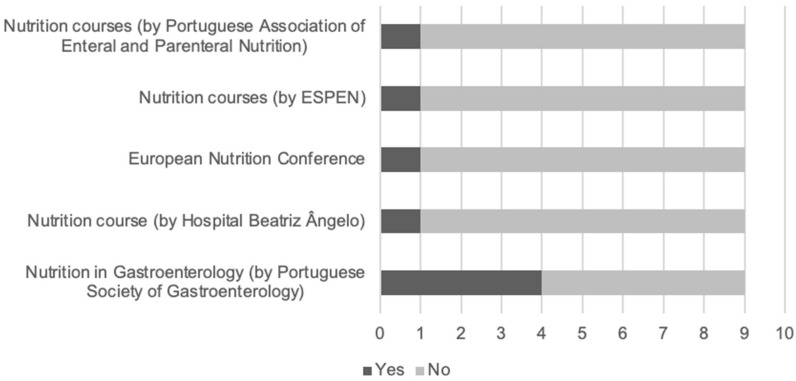
Nutrition-related courses by number of departments.

**Figure 2 ijerph-19-16333-f002:**
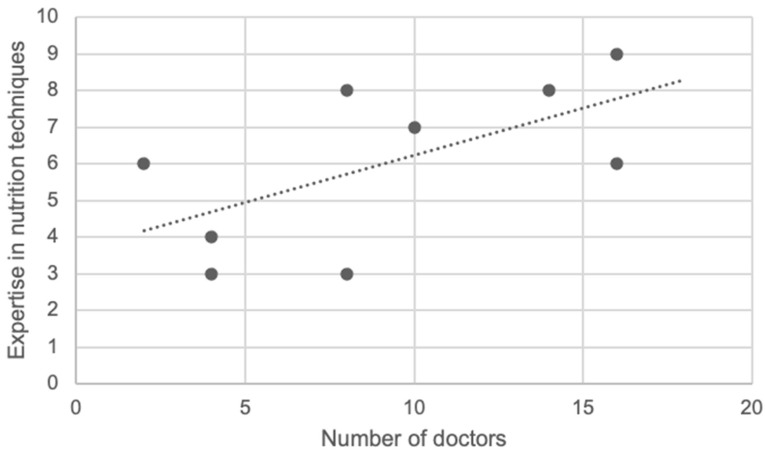
Linear correlation between number of doctors and expertise in nutrition techniques. Black dots refer to data from each involved center (*n* = 9).

**Table 1 ijerph-19-16333-t001:** Department characterization in 2018.

	Population	Ward (Beds)	Gastro Doctors	Ratio Patient/ Doctor	Type of Outpatient Clinic					
					AN	IBD	Hepatology	Proctology	Oncology	Pancreatic Diseases
A	150,000	No	2	75,000	No	Yes	Yes	No	No	No
B	120,000	Yes (10)	10	12,000	No	Yes	Yes	Yes	No	Yes
C	150,000	No	8	18,750	Yes	Yes	Yes	Yes	No	No
D	350,000	Yes (16)	16	21,975	No	Yes	Yes	Yes	Yes	No
E	295,000	No	4	73,750	No	Yes	Yes	Yes	No	No
F	130,000	Yes (12)	8	16,250	No	Yes	Yes	Yes	No	No
G	350,000	Yes (12)	14	25,000	Yes	Yes	Yes	Yes	Yes	No
H	380,000	No	4	95,000	No	Yes	No	No	Yes	No
I	278,000	Yes (15)	16	17,375	No	Yes	Yes	Yes	Yes	Yes

AN–artificial nutrition; IBD–inflammatory bowel disease.

**Table 2 ijerph-19-16333-t002:** Nutrition techniques by each department in 2018.

	Enteral Nutrition					Parenteral Nutrition			
	NGT	NET	PEG	PEG-J	PEJ	PVC	PICC	CVC	T-CVC
A	Y (0)	Y (10)	Y (30)	Y (0)	N	Y (0)	N	Y (0)	N
B	Y (10)	Y (8)	Y (24)	N	N	Y (0)	Y (0)	Y (0)	Y (0)
C	Y (38)	Y (25)	Y (44)	Y (0)	N	Y (0)	Y (0)	Y (0)	Y (1)
D	Y (50)	Y (10)	Y (20)	Y (0)	Y (0)	Y (20)	N	N	N
E	Y (2)	Y (1)	Y (15)	N	N	N	N	N	N
F	Y (0)	Y (0)	Y (34)	N	N	N	N	N	N
G	Y (4)	Y (11)	Y (82)	Y (1)	Y (0)	Y (14)	N	Y (25)	Y (4)
H	Y (15)	Y (22)	Y (12)	N	N	N	N	Y (0)	N
I	Y (5)	Y (12)	Y (26)	Y (0)	Y (0)	Y (7)	Y (2)	Y (12)	Y (3)

Y-expertise with number of techniques performed in 2018 in parentheses. N, no expertise; NGT, nasogastric tube; NET, nasoenteric tube; PEG, percutaneous endoscopic gastrostomy; PEG-J, percutaneous endoscopic transgastric jejunostomy; PEJ, percutaneous endoscopic jejunostomy; PVC, peripheral venous catheter; PICC, peripherally inserted central catheter; CVC, central venous catheter; T-CVC, tunneled central venous catheter.

## Data Availability

The datasets analyzed during the current study are available from the corresponding author upon reasonable request.

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
