# Peer review of "Clinical Nutrition in Portuguese Gastroenterology Departments: A Multicentric Study"

_ijerph, 2022, doi:10.3390/ijerph192316333_

Round 1
Reviewer 1 Report
As detailed in the attached review the paper needs to be revised but brings relevant attention to the need for development of the clinical nutrition part of gastroenterology in general.

Author Response
The authors thank the reviewers’ work, which helped to strengthen this manuscript.
We tried to address all comments point-by-point:
- Title: We have deleted the words “nationwide” and “artificial”, but we have maintained “multicentric study”, as it describes the design of our study.
- Abstract: The authors acknowledge that “50% of hospitalized patients suffer from undernutrition”, is too categoric and generalized. We believe adding the words “up to 50%” is a better way to characterize the undernutrition’s burden. As in the title, “nationwide” was deleted. We, also, mentioned that the questionnaire was sent to 31 departments. The concept of multidisciplinary team on that sentence arises from a unit with a doctor, nurse and a dietitian, who together take care of multiple complex cases. The authors deleted that word in the abstract as “multidisciplinary” appears below in the manuscript with another meaning (point 6 of this reply).
- Introduction: The authors added information regarding the impact of nutrition in liver diseases and in IBD patients. Also, the sentence “Both in health and disease, human nutrition results from the complex interplay between the digestive system and the brain” was changed to try not to oversimplify this concept.
- Materials and Methods: The authors added the information that one reminder was sent to the nonresponders.
- Discussion: The authors tried to be more concrete on the discussion, with simple suggestions that may improve the nutrition care in gastroenterology departments.
- Conclusion: The multidisciplinary management of nutrition-related disorders is referred to highlight the importance of different departments’ involvement in these patients (for example internal medicine, general surgery), as in the hospital nutrition committee.
Reviewer 2 Report
1.Although it doesn't reflect the entire complexity of patient nutrition, I think that is a good study to start with, specially if you want to raise the awareness.
2. More focus should be added on personalized nutrition (please add something about this in section 4 :Discussion), as the spectrum of patients is different in this centers and some centers treat also patients with GI cancers,and nutritional status is extremely important for this patients in the course of chemotherapy, please add something about the needs of patients with different pathology - some need PEG, PEG-j etc, some need enteral stents, biliary stents, pancreatic stents, duodenal stents, some need PTCD drainage, just to add up on the real need of interventional endoscopy training of trainees and doctors that can impact the nutritional status.
"Pancreatic cancer Malnutrition and pancreatic Exocrine insufficiency in the course of chemotherapy in unresectable pancreatic cancer" Mariia Kiriukova et al. DOI: 10.3389/fmed.2020.00495
Some data about enteral stenting would have been helpful ( how many doctors can perform enteral stenting, ERCP( with stent insertion in billiary and pancreatic malignancies or for patients with chronic pancreatitis), EUS (EUS-GJ with LAMS or main pancreatic duct drainage etc.) but all this can be the focus of future research as its is difficult to add them now.
3. Some data about the experience of other countries can be helpful to add
"Improving 30 day mortality after PEG tube placement in England from 2007 to 2019: a retrospective national cohort analysis of 87862 patients"Umair Kamran et al. GIE ( December 2022)
Author Response
The authors thank the reviewers’ work, which helped to strengthen this manuscript.
We tried to address all comments point-by-point:
- The authors added information regarding personalized nutrition in some contexts, for example cancer patients, including the need for different nutrition technique according to the clinical scenario.
- The authors added information on the final paragraph of the discussion regarding the need to further study the actual state of enteral stenting, ERCP and EUS for nutrition purposes.
- The authors added data of other countries as suggested.